# Antiviral Effect of Melatonin on Caco-2 Cell Organoid Culture: Trick or Treat?

**DOI:** 10.3390/ijms252211872

**Published:** 2024-11-05

**Authors:** Milda Šeškutė, Dominyka Žukaitė, Goda Laucaitytė, Rūta Inčiūraitė, Mantas Malinauskas, Lina Jankauskaitė

**Affiliations:** 1Department of Pediatrics, Lithuanian University of Health Sciences, 44307 Kaunas, Lithuania; milda.seskute@lsmu.lt (M.Š.); goda.laucaityte@lsmu.lt (G.L.); 2Institute of Physiology and Pharmacology, Lithuanian University of Health Sciences, 44307 Kaunas, Lithuania; mantas.malinauskas@lsmu.lt; 3Faculty of Medicine, Lithuanian University of Health Sciences, 44307 Kaunas, Lithuania; dominyka.zukaite@stud.lsmu.lt; 4Institute for Digestive Research, Lithuanian University of Health Sciences, 44307 Kaunas, Lithuania; ruta.inciuraite@lsmu.lt

**Keywords:** melatonin, gastrointestinal organoids, Caco-2 cell, antiviral effect, IFNLR1, IFNλ1

## Abstract

Melatonin is a hormone naturally produced by the body that has recently been found to have antiviral properties. However, its antiviral mechanisms are not entirely understood. Using Caco-2 cells, we developed a gastrointestinal organoid model to investigate the impact of melatonin on cellular organoid culture response to Poly I:C-induced viral inflammation in the gastrointestinal tract. Melatonin was found to have different effect when applied as a pretreatment before the induction of viral inflammation or as a treatment after it. Melatonin pretreatment after Poly I:C stimulation did not protect organoids from size reduction but enhanced cell proliferation, especially when lower (1 and 10 µM) melatonin concentrations were used. On the other hand, treatment with melatonin after the induction of viral inflammation helped to maintain the size of the organoids while reducing cell proliferation. In pretreated cells, reduced IFNLR1 expression was found, while melatonin treatment increased IFNLR1 expression and reduced the production of viral cytokines, such as IFNλ1 and STAT1-3, but did not prevent from apoptosis. The findings of this study emphasize the importance of type III IFNs in antiviral defense in epithelial gastrointestinal cells and shed more light on the antiviral properties of melatonin as a potential therapeutic substance.

## 1. Introduction

Viral gastrointestinal infections are a major public health concern worldwide causing significant morbidity and mortality, especially in small children and elderly people [1]. Currently, the management for these infections is only symptomatic (e.g., infusion therapy, antipyretics, etc.). Thus, there is a vital necessity for a single pan-antiviral therapeutic agent [1]. Recent evidence suggests melatonin has an antiviral effect that could benefit in combating gastrointestinal diseases [2,3,4].

Melatonin is commonly known as a hormone produced by the pineal gland to control the sleep–wake cycle [5]. However, it has numerous other beneficial effects, and due to its strong antioxidative characteristics, it recently became widely investigated as a possible therapeutic agent for various non-infectious and infectious diseases [4,5,6,7,8]. The antiviral effect of melatonin was first detected with encephalomyocarditis virus (ECMV) in rodents in 1988 [9]. Later, this effect was confirmed by investigating other viral infections. Melatonin has already been shown to inhibit the replication of viruses, such as severe acute respiratory syndrome coronavirus 2 (SARS-CoV-2), Zika, Japanese encephalitis virus, and herpesvirus type I, among others [6,8,10,11,12]. The antiviral effect of melatonin is primarily attributed to the reduction in apoptosis and the increase in autophagy through the reduction of oxidative stress [12]. However, melatonin is also debated to have anti-inflammatory effects, as well as immunoenhancing properties that boost humoral and/or cellular immune responses [8]. Melatonin plays an important role in the proliferation and maturation of various immune cells, such as T and B cells, and decreases the production of pro-inflammatory mediators (interleukin (IL)-2, IL-6, IL-8, interferon (IFN)-α, tumor necrosis factor (TNF)-α, nuclear factor kappa B (NFκB), etc.) [2,3], but knowledge about antiviral mechanisms, especially in the gastrointestinal tract, is still scarce.

Recent animal studies have shown that melatonin in the gut is produced independently from the pineal gland, which shows its importance in this organ system [13,14]. Gastrointestinal melatonin regulates electrolyte and ion transport, gastrointestinal motility, and epithelial regeneration and has gastroprotective properties [13,15]. Recent advances in gastrointestinal organoid development [16,17] have also allowed more precise research of human processes without having to use animal models. In our research, we developed a gastrointestinal organoid model to investigate the effects of melatonin on organoids and the immune response to viral inflammation. We analyzed the organoid size, proliferation, cell apoptosis and interferons IFNα, IFNβ, and IFNλ1 production as well as the subsequent STAT1-3 (signal transducer and activator of transcription 1–3) protein expression. We found that melatonin influences cell viability, organoid formation, interferon lambda (IFNλ), and other components of the type III IFN pathway and may have different effects when applied as a pretreatment before the induction of viral damage or as a treatment after it.

## 2. Results

### 2.1. Development of Caco-2 Cell Viral Model

Caco-2 cells were first cultured as a monolayer for 21 days, leading to the formation of organoids (Figure 1A–G). The impact of Poly I:C (polyinosinic polycytidylic acid) on cell viability, as well as the number and size of the resulting organoids, was subsequently assessed. Varying concentrations (1, 10 or 50 µg/mL) of Poly I:C were used to induce viral damage. A concentration-dependent decrease in cell viability was observed with higher concentrations of Poly I:C (Figure 2A). Also, both the number and size of the gastrointestinal organoids were significantly reduced at higher concentrations of Poly I:C (Figure 2A,C–E).

To optimize conditions for a viral model, the release of inflammatory cytokines, including IFNλ1, IFNβ, and IFNα, was analyzed. Poly I:C stimulation induced a concentration-dependent production of IFNλ1 in Caco-2 cells (Figure 3A), with a significant increase in IFNλ1 levels detected after at least 6 h of Poly I:C stimulation at all concentrations tested (Figure 3B). In contrast, no detectable expression of IFNβ or IFNα was observed under the same conditions (Figure 3C,D).

### 2.2. The Effect of Melatonin on Poly I:C-Stimulated Caco-2 Cells

Afterwards, we analyzed the effect of melatonin on Caco-2 cells. Before the application on the viral model, the effect of melatonin was tested on non-stimulated Caco-2 cells. Varying concentrations of melatonin did not significantly impact Caco-2 cell viability (Figure 2B) nor alter the characteristics of the gastrointestinal organoids (Figure 2C–E).

When applied prior to Poly I:C stimulation, pretreatment with melatonin enhanced Caco-2 cell proliferation (Figure 4A); however, this proliferation-promoting effect was not sustained at higher melatonin concentrations (50 and 100 μM). In contrast, post-stimulation treatment with melatonin did not preserve cell viability (Figure 4A).

The melatonin pretreatment failed to protect organoids from the damage of Poly I:C. Although the number of organoids remained relatively stable following the administration of 50 µg/mL Poly I:C, a reduction in organoid size was observed (Figure 4B,C), mirroring the control group. Conversely, melatonin treatment after Poly I:C stimulation, particularly at lower concentrations, resulted in a reduction in the number of organoids, but those that remained exhibited an increase in size, suggesting a potential protective effect (Figure 4B,C).

Melatonin pretreatment showed a tendency to protect Poly I:C-stimulated cells from apoptosis, particularly at low doses (1 and 10 μM) (Figure 4D,E). However, at higher melatonin concentrations administered post Poly I:C stimulation, apoptosis levels were comparable to those observed in the melatonin-naïve group.

### 2.3. Melatonin Decreases IFNλ1 Levels in Poly I:C-Stimulated Caco-2 Cells When Applied as a Treatment

In contrast to Poly I:C-stimulated cells, melatonin did not induce the production of IFNλ1 in non-infected cells (Figure 5A). When melatonin was applied as a pretreatment before Poly I:C stimulation, IFNλ1 levels were similar to those observed in melatonin-naive cells. However, treatment with melatonin following Poly I:C stimulation resulted in reduced levels of IFNλ1 (Figure 5A). Specifically, melatonin treatment after stimulation with low concentrations of Poly I:C (1 and 10 μM) led to IFNλ1 levels that were nearly indistinguishable from the control cells. Across different concentrations, melatonin exhibited a similar protective effect.

### 2.4. Melatonin Treatment Increases IFNLR1 Expression in Caco-2 Cells

To further evaluate the antiviral effect of melatonin via the type III IFN pathway, we analyzed the expression of the IFNλ receptor 1 (IFNLR1). Elevated levels of IFNLR1 were observed to be expressed in response to Poly I:C stimulation (Figure 5B,C). Notably, melatonin also induced IFNLR1 expression in non-infected cells. Treatment with melatonin further increased IFNLR1 expression in Poly I:C-stimulated cells, while pretreated ones exhibited lower amounts of IFNLR1 (Figure 5B). No significant differences were detected between varying concentrations of melatonin.

In the pretreated cells, an inverse correlation was observed between the expression of IFNλ and IFNLR1 (r = −0.475, *p* = 0.03), whereas no significant correlation was found between the two markers in melatonin-treated cells post Poly I:C stimulation (*p* = 0.528).

### 2.5. Melatonin Treatment Reduces the Expression of STAT Proteins

The expression of STAT1, STAT2, and STAT3 were subsequently analyzed to evaluate melatonin’s effect on the Janus kinase/signal transducer and activator of transcription (JAK/STAT) pathway and its possible relationship with IFNλ1 in gastrointestinal cells. Poly I:C stimulation significantly upregulated STAT1-3 expression in Caco-2 cells compared to the control cells (Figure 6A–C). Melatonin pretreatment reduced STAT1 expression in infected Caco-2 cells while non-infected melatonin-pretreated Caco-2 cells exhibited higher levels of STAT1 compared to the controls (Figure 6A). STAT2 and STAT3 levels remained similar in melatonin-pretreated cells (Figure 6B,C). Treatment with melatonin following Poly I:C stimulation resulted in a reduction in the expression of all three STAT proteins, with no significant differences across varying melatonin concentrations.

A moderate correlation was observed among the STAT proteins, with the strongest association between STAT1 and STAT3 (r = 0.683, *p* < 0.001). The levels of STAT1-3 did not significantly correlate with apoptosis in either pretreated or treated cells.

In melatonin-pretreated cells, IFNλ1 levels showed a moderate correlation with STAT1-3 expression, with the highest correlation observed for STAT2 (r = 0.509). STAT2 and STAT3 also moderately correlated with IFNLR1 (r = 0.506 and r = 0.438, respectively). In melatonin-treated cells, there was a weak or inverse correlation between STAT1-3 and IFNλ1 and no significant correlation between IFNLR1 and STAT proteins.

## 3. Discussion

In this study, we explored the antiviral effect of melatonin in a Caco-2 cell two-dimensional model. Our findings revealed distinct responses in infected cells, depending on whether melatonin was applied as pretreatment or treatment. Pretreatment with low concentrations of melatonin enhanced cell proliferation and reduced apoptosis, whereas post-treatment preserved the size of organoids, increased IFNLR1 expression, and reduced antiviral cytokines production (IFNλ1 and STAT1-3) in response to Toll-like receptor (TLR)-3 stimulation imitating viral infection, suggesting possible protective properties (Figure 7).

While melatonin has been studied in the treatment of various viral infections [4,6,8,10,11,12], studies focusing on gastrointestinal viral infections remain limited. Xi-Zhang et al. [2] demonstrated that melatonin reduces inflammation caused by bacterial pathogens in gastrointestinal organoids by lowering the expression of NFκB, IL-6 and IL-8. In our study, we explored the antiviral potential of melatonin by investigating the expression of various types of interferons, which play a major role in the antiviral response.

Historically, type I IFNs were considered central to antiviral defense. However, the discovery of type III IFNs in 2003 revealed their essential role in protecting various epithelial surfaces, including intestinal epithelial cells, from viral damage [18,19]. Although it is not fully understood which specific intestinal epithelial cells produce IFNλ [18], our findings showed that differentiated human Caco-2 cells produced IFNλ1 in response to TLR3 stimulation. This aligns with other animal and human studies which showed preferential IFNλ production over type I IFNs in intestinal cells during enteric infections [19]. Interestingly, we observed no production of IFNβ and IFNα in our experiments, partially contradicting the results of Frias et al. [20], who reported that gastrointestinal epithelial cells produce IFNβ in response to rotavirus infection and induce cell death to possibly clear the virus. The discrepancies may be attributable to differences in the intestinal cells and methodologies used across studies.

Despite extensive research on melatonin as a potential antiviral agent, its effect on IFNλ is not well characterized. Our study showed that melatonin treatment following TLR3 stimulation reduced IFNλ1 levels. Since IFNλ is known to inhibit cell proliferation, this effect may be related to upregulation of IFNLR1 [21]. While we were unable to confirm a direct link, we found that melatonin treatment reduced IFNλ1 expression and cell proliferation. In contrast, melatonin pretreatment enhanced Caco-2 cell proliferation but without significantly affecting IFNλ1 levels, suggesting that alternative pathways might mediate the proliferation-promoting effect of melatonin. Interestingly, we observed an inverse correlation between IFNλ1 and IFNLR1 expression in pretreated cells, while no such relationship was found in the treatment group, implying the involvement of additional factors.

Previous studies have concluded that IFNλ1 inhibits cell proliferation via IFNLR1 activation and the subsequent tyrosine phosphorylation of STAT proteins [22]. Animal models also have shown that the overexpression of IFNLR1 activates the JAK/STAT pathway [21]. However, the molecular mechanisms underlying this pathway remain poorly understood. In our study, we examined STAT protein expression and found that melatonin alone induced STAT2 and STAT3 expression even in the absence of viral damage despite lower IFNLR1 levels compared to control cells. In melatonin-pretreated Poly I:C-stimulated cells, we observed a correlation between IFNLR1 and STAT1-3 as reported in other studies [21], while an inverse but statistically insignificant correlation was found in the treatment group.

Various animal models have already suggested the effect of melatonin on STAT proteins. Melatonin inhibited the activity of STAT1 through the suppression of nitric oxide and IL-6 production in the context of inflammation induced by bacterial lipopolysaccharide (LPS) in murine macrophages [23]. Melatonin pretreatment was also found to suppress the expression of STAT1 but activate STAT3 phosphorylation in microglia cells in the case of neuroinflammation [24]. Additionally, melatonin was recently shown to directly inhibit the phosphorylation of STAT3 and lower cellular senescence and apoptosis in the case of diabetic kidney injury [25]. However, its effect on STAT proteins in the gastrointestinal tract is not well described. Also, we could not find any studies that explored melatonin’s role on STAT2 expression, while our research showed STAT2 to have strongest correlation with IFNλ1 expression. STAT2 might play a key role in antiviral response to certain viruses and may independently from other STAT proteins promote interferon-stimulated gene (ISG) production [26].

In our study, IFNλ1 or IFNLR1 expression was not influenced by different melatonin concentrations. However, we observed distinct dynamics of these cytokines and their receptor in pretreated and treated cells. Winkle et al. suggested that IFNλ signaling in rotaviral infection models may have a prophylactic role [18], showing that IFNLR1 limits the viral evasion of intestinal epithelial cells. Interestingly, in our experiments, melatonin-treated cells exhibit higher IFNLR1 expression than pretreated cells.

While melatonin is generally regarded as safe for short-term use even at relatively high doses [27], data about its long-term safety are limited [28]. Animal studies have raised concerns that melatonin might adversely impact fertility [29]. Additionally, different doses of melatonin have been studied for various disease contexts, and there remains no established consensus on the optimal dosage or timing for its use in infection prevention or treatment. Maestroni et al. hypothesized [8] that higher doses of melatonin may be required before the infection to stimulate the innate immune response, while lower doses post infection may reinforce adaptive immunity [8]. In our experiments, we used different melatonin concentrations at varying timepoints (before and after the induction of viral damage) and found that the effect of melatonin might depend on the dosage and timing. Lower melatonin concentrations prior to Poly I:C stimulation reduced Caco-2 cell viability, while higher concentrations or post-stimulation treatments did not sustain this effect. This is consistent with other studies suggesting that melatonin promotes epithelial regeneration at lower doses, while higher doses may have an inhibitory effect [15]. Moreover, studies with cancer cells have revealed that melatonin application on cancer cells causes dose-dependent cell toxicity and cellular damage with a decrease in cell viability by at least 40% when 4 mM or higher doses of melatonin were used [30,31]. However, the effect on non-cancerous cells might be different. Therefore, dosage emerges as a critical variable for future preclinical and clinical studies. Optimizing the concentration of melatonin is essential to balance its protective effects while minimizing potential cytotoxicity or adverse outcomes. Moreover, the timing of melatonin administration warrants careful consideration. Early administration, which is associated with reduced antiviral protein expression, may inadvertently exacerbate viral inflammation or induce cellular apoptosis. These findings underscore the importance of delineating the precise therapeutic window for melatonin treatment, where its immunomodulatory benefits are maximized without triggering detrimental side effects. This highlights the need for detailed pharmacokinetic and pharmacodynamic studies to guide its clinical use.

Our study has several limitations. First, we used monolayer culture consisting solely of epithelial intestinal cells. While this method is commonly used, the cellular response may differ and be better applicable in vivo, particularly if additional cell types, such as fibroblasts and endothelial cells, are incorporated to more closely mimic the human intestinal tract. Additionally, more accurate data could be reached by using a three-dimensional culture model instead of a monolayer. We are currently working to refine our model for future experiments. Second, we stimulated viral inflammation using a TLR3 receptor agonist rather than live virus infection due to our laboratory safety limitations. Our experiments on stimulating the TLR3 and TLR7/8 pathways resulted in increased intestinal organoid sequestration and those cultures could not be used for the further experiments. The effect of melatonin could differ in the context of live virus infection due to additional host–pathogen interactions.

Nevertheless, our results are consistent with some animal studies, showing that our model provides meaningful insights that can be validated in future studies. Finally, we analyzed only some subsets of the type III IFN pathway, and additional studies, including the investigation of IFNλ1 stimulated genes, are necessary to elucidate the full mechanism of melatonin’s action in the gastrointestinal tract.

## 4. Materials and Methods

### 4.1. Cell Culture and Caco-2 Organoid Development

Caco-2 cells were obtained from ATCC (United Kingdom (UK)) and cultivated in cell culture flasks in minimum essential medium (MEM; Gibco, Life Technologies Limited, Paisley, UK) + 10% fetal bovine serum (FBS; Gibco, Life Technologies NZ Ltd., Auckland New Zealand) + 1% non-essential amino acids (Gibco, Life Technologies Limited, Paisley, UK) + 1% penicillin/streptomycin (Gibco, Life Technologies Limited, Paisley, UK) at 37 °C in a humidified incubator at 5% carbon dioxide (CO_2_). The cells were passaged upon reaching 70–80% confluence. Passages 9 to 18 were used for the experiments. For all the experiments, the cells were harvested using TrypLE™ Express Enzyme (Gibco, Life Technologies Corporation, Grand Island, NY, USA), counted using a hemacytometer and Trypan Blue 0.4% stain (Gibco, Life Technologies Corporation, Grand Island, NY, USA), and seeded at a density of 1 × 10^4^ cells/well into 96-well tissue culture plates. The cells were cultured with media changes every 2–3 days as a monolayer for 21 days to form organoids. The outline of further experiments is shown in Figure 8.

### 4.2. Stimulation with Poly I:C

Caco-2 cells, cultured in 96-well plates for 21 days, were stimulated for 24 h at concentrations of 1, 10 or 50 µg/mL of TLR3 agonist polyinosinic polycytidylic acid (Poly I:C; Tocris Bioscience, Abingdon, UK).

### 4.3. Melatonin Treatment

Melatonin was prepared according to the manufacturer’s instructions (Sigma-Aldrich, St. Louis, MO, USA) and subsequently diluted in MEM with supplements as previously described. For the experiment procedure, Caco-2 cells were either pretreated with melatonin at concentrations of 1, 10, 50 or 100 µM 24 h prior to Poly I:C stimulation or treated with melatonin of 1, 10, 50 or 100 µM 24 h following Poly I:C stimulation (Figure 8).

### 4.4. Measurement of Organoid Number and Size

To compare the effect of melatonin pretreatment and treatment on Poly I:C stimulation on the organoid growth and morphology, Caco-2 organoids were imaged at 24 h post Poly I:C stimulation or 24 h post melatonin application (in the pretreatment group) in 96-well plates. Brightfield images of organoid cultures were captured using an inverted microscope (Olympus IX2-SP, Olympus Corporation, Tokyo, Japan) equipped with a digital camera (Olympus DP26, Olympus Corporation, Tokyo, Japan). All images were taken at consistent magnification (4×) for accurate comparisons. The number and size of Caco-2 organoids were measured using ImageJ software (version 1.53k, National Institutes of Health, Bethesda, MD, USA). The number of organoids was evaluated by manually counting each visible organoid per image using ImageJ. For each condition, the mean organoid size was calculated. To ensure consistency, the same investigator performed all image analyses under identical conditions. The analysis was performed in triplicates of 3 different wells per condition.

### 4.5. Staining of Caco-2 Organoid Markers

Caco-2 cell cultures with organoids were washed with phosphate-buffered saline (PBS, Gibco, Life Technologies Limited, Paisley, UK) and subsequently fixed with 4% paraformaldehyde (PFA) for 15 min at room temperature. After rinsing with PBS, the cells were permeabilized with 0.2% Triton™ X-100 (Sigma-Aldrich, St. Louis, MO, USA) for 1 h at room temperature to facilitate antibody penetration for intracellular markers (ZO-1 and vimentin). Cells were then blocked with 10% FBS in PBS for 1 h at room temperature. Following the aspiration of the blocking buffer, the cells were incubated overnight at 4 °C with the human monoclonal antibodies to target the specific organoid markers: CD326 (EpCAM, epithelial cell adhesion molecule) antibody (1:1000, Invitrogen, Carlsbad, CA, USA) for epithelial cell identification; ZO-1 (zonula occludens-1) fluorescein isothiocyanate (FITC) antibody (1:1000, Invitrogen, Rockford, IL, USA) for tight junction visualizations; and vimentin antibody (1:1000, Invitrogen, Rockford, IL, USA) for the cells of mesenchymal origin. After antibody incubation, cultures were washed with PBS and counterstained with DAPI (Sigma-Aldrich, St. Louis, MO, USA) for 10 min at room temperature to stain nuclei. After staining, the organoids prepared in 96-well plates and were imaged using an inverted microscope equipped with a digital camera and an X-cite 120Q fluorescence illuminator (Lumen Dynamics, ON, Canada). Images were acquired with cellSens software (version 1.16, Olympus, Olympus Corporation, Japan) and processed using ImageJ software. All incubation steps were performed in the dark.

### 4.6. Cell Viability Assays

Cell viability was assessed using Cell Counting Kit-8 (WST-8/ CCK8; Abcam, Cambridge, UK) at 24 and 48 h post Poly I:C stimulation. As instructed by the manufacturer, 10 μL of the CCK-8 solution was added to each well. Following a 4 h incubation period, the absorbance was measured at 450 nm wavelength using a microplate reader (Thermo Fisher Scientific, Waltham, MA, USA).

### 4.7. ELISA Assays

The levels of antiviral cytokines (IFNλ1, IFNβ, and IFNα) in the cell culture supernatant were quantified using commercially available human enzyme-linked immunosorbent assay (ELISA) kits (IFNλ1, Abcam, Cambridge, UK, detection limit—13.72 ng/mL; IFNβ, Abcam, Cambridge, UK, detection limit—9.38 pg/mL; and IFNα, Invitrogen, Bender MedSystems GmbH, Vienna, Austria, detection limit—7.8 pg/mL). Supernatants were harvested and stored at −80 °C before the measurement. According to the manufacturer’s instructions, the analyzed cytokines were incubated with specific monoclonal antibodies and subsequently with horseradish peroxidase. The resulting optical density (OD) values were measured and converted into concentration values. All assays were performed in triplicate for statistical robustness.

### 4.8. Flow Cytometry

Multicolor flow cytometry analysis (FACS) was performed with a BD FACSMelody cell sorter using BD FACSChorus software (version 3.0, Becton, Dickinson and Company, Franklin Lakes, NJ, USA). For FACS, 1–5 × 10^5^ cells were resuspended in 2% FBS + PBS solution. To assess STAT1-3 and IFNLR1 expression, cells were stained with specific antibodies as follows: Phospho-STAT1 (Tyr701) monoclonal antibody eFluor™ 660 (eBioscience™, Life Technologies Corporation, Carlsbad, CA, USA), Phospho-STAT2 (Tyr689) polyclonal antibody, FITC (Invitrogen, Rockford, IL, USA), Phospho-STAT3 (Tyr705) monoclonal antibody, phycoerythrin (PE) (eBioscience™, Life Technologies Corporation, Carlsbad, CA, USA) and IFNLR1 monoclonal antibody (Invitrogen, Rockford, IL, USA). All staining procedures were conducted according to the manufacturer’s instructions. The cells were washed in PBS, trypsinized using TrypLE™ Express Enzyme. For intracellular staining, the cells were permeabilized with 0.2% Triton™ X-100 solution for 10–15 min followed by incubation with fluorescent-labeled secondary antibodies (Phospho-STAT1 (Tyr701), Phospho-STAT2 (Tyr689), Phospho-STAT3 (Tyr705)) for at least 30 min at 4 °C. For surface marker IFNLR1 staining, the cells were incubated with nonlabelled primary IFNLR1 monoclonal antibody overnight at 4 °C followed by incubation with an appropriate fluorescent-labeled secondary antibody (Alexa Fluor™ 488, Invitrogen, Eugene, OR, USA) for at least 30 min at 4 °C the next day and analyzed using a flow cytometer. Corresponding isotype antibodies and unstained cells were used as negative controls. The results were analyzed using FlowJo 10.10 software (Becton, Dickinson and Company, Franklin Lakes, NJ, USA). Median fluorescence intensity (MFI) values were used to quantify the expression levels of the analyzed antibodies. The percentage of cells positive for each marker was calculated by setting a threshold based on the control.

### 4.9. Assessment of Apoptosis

Apoptosis was assessed using the Annexin V/propidium iodide kit (Invitrogen, Life Technologies Corporation, Eugene, OR, USA). The cells were harvested using TrypLE™ Express Enzyme and stained with Annexin V/propidium iodide for at least 15 min and immediately submitted for analysis by a flow cytometer. All assays were conducted in triplicates.

### 4.10. Statistical Analysis

Statistical analysis was performed using IBM SPSS Statistics (version 29.0.2.0, IBM Corp., Armonk, NY, USA) and GraphPad Prism software (version 10.3.1, GraphPad Software, Boston, MA, USA). The results are displayed as mean +/− standard deviation (SD) of at least three independent experiments. Statistical significance is analyzed by unpaired Student’s *t*-test. Analysis of variance (ANOVA) and post hoc Turkey were used for multiple-group analysis. Correlations were counted using the Pearson coefficient. Statistical significance was assumed when the *p*-value was less than 0.05 and indicated as * *p* < 0.05, ** *p* < 0.01, and *** *p* < 0.001. GraphPad Prism software was used to draw the graphs. Schemes were created using Biorender.com (accessed on 19 July 2024).

## 5. Conclusions

Our findings demonstrate that melatonin induces an antiviral effect in gastrointestinal epithelial cells via the type III IFN pathway, with distinct outcomes depending on whether it is used as a treatment after the onset of viral infection or as a preventive treatment before it. When administered as a treatment, melatonin displayed protective properties by preserving organoid size and upregulating IFNLR1 expression and weakening antiviral cytokine responses, including a reduced expression of IFNλ1 and STAT1-3. Although it did not prevent cellular apoptosis, the observed effects on cytokines suggest that melatonin may help balance the antiviral response to limit tissue damage. In contrast, pretreatment triggered different protective mechanisms where IFNLR1 expression was reduced but cytokine production remained responsive to inflammation, promoting cell proliferation and anti-apoptotic effects, especially at lower concentrations, indicating a potential role for melatonin in preparing cells for a viral challenge.

These results underscore the critical role of type III IFNs in antiviral defense as well as dual antiviral and protective qualities of melatonin in gastrointestinal epithelial cells, highlighting its therapeutic potential in managing viral intestinal diseases. However, our findings emphasize the need for a careful consideration of melatonin’s timing and dosage in clinical applications. Further research is needed in more complex organoid models to clarify melatonin’s antiviral mechanisms and its effect across different cell types.

## Figures and Tables

**Figure 1 ijms-25-11872-f001:**
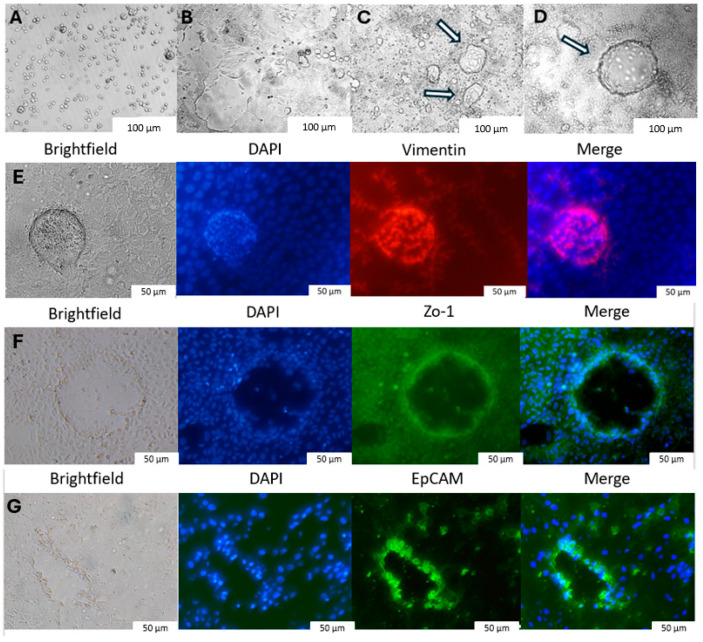
Brightfield microscopy of Caco-2 cell culture (**A**) Day 0, (**B**) Day 7, (**C**) Day 14, (**D**) Day 21 showing the formation of gastrointestinal organoids (indicated by arrows). Brightfield and immunohistochemistry images of Caco-2 showing different markers: (**E**) vimentin, (**F**) Zo-1 (zonula occludens-1), and (**G**) EpCAM (epithelial cell adhesion molecule).

**Figure 2 ijms-25-11872-f002:**
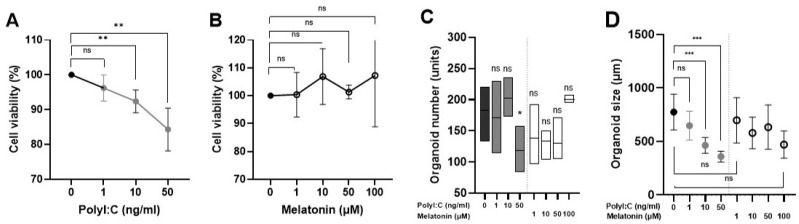
(**A**) Caco-2 cell viability after stimulation with different concentrations of Poly I:C for 24 h (n = 5). (**B**) Caco-2 cell viability after stimulation with different concentrations of melatonin for 24 h (n = 3). (**C**) Organoid size after stimulation with different concentrations of Poly I:C or melatonin for 24 h (means with 95% confidence interval (CI)). (**D**) Number of organoids after stimulation with different concentrations of Poly I:C or melatonin for 24 h compared to control (means with min and max values). (**E**) Brightfield images of Caco-2 cells monolayer culture after stimulation with different concentrations of Poly I:C or melatonin for 24 h alone in comparison to stimulation with 1 and 100 μM of melatonin before (pretreatment) and after (treatment) application of 1, 10 and 50 µg/mL Poly I:C; reducing size and number of organoids when stimulated with higher Poly I:C concentrations are seen, while higher number of organoids when pretreated with melatonin and stimulated with high Poly I:C concentrations in comparison to control cells are observed. * indicates *p* < 0.05, ** indicates *p* < 0.01, *** indicates *p* < 0.001, and ns indicates *p* > 0.05. Abbreviations: µg/mL—micrograms per milliliter, µm—micrometer, µM—micromolar, Poly I:C — polyinosinic polycytidylic acid.

**Figure 3 ijms-25-11872-f003:**
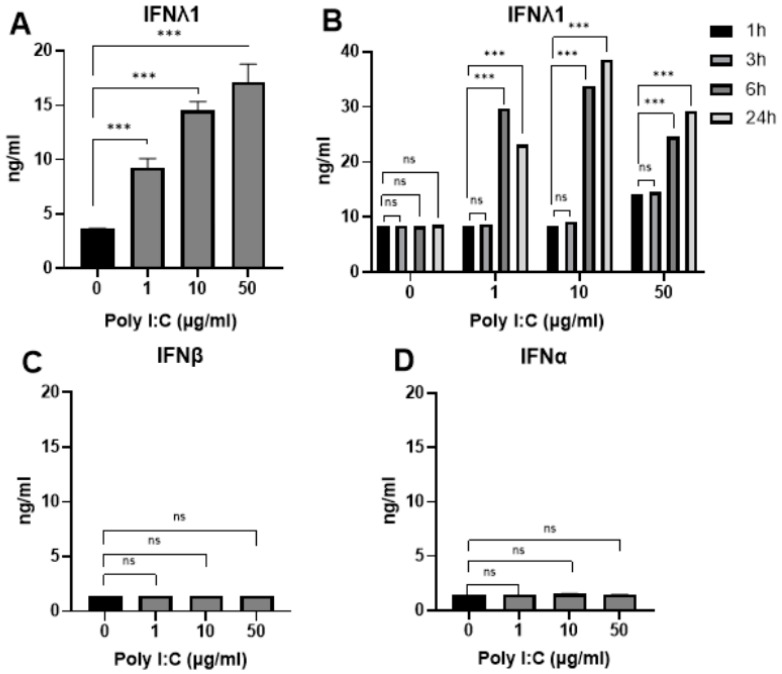
The levels of interferons produced by Poly I:C-stimulated Caco-2 cells detected by ELISA. (**A**) The levels of IFNλ1 after stimulation of Caco-2 cells with different concentrations of Poly I:C after 24 h (n = 3). (**B**) The levels of IFNλ1 at different timepoints after stimulation with Poly I:C (n = 3). (**C**) The levels of IFNβ after stimulation of Caco-2 cells with different concentrations of Poly I:C after 24 h (n = 3). (**D**) The levels of IFNα after stimulation of Caco-2 cells with different concentrations of Poly I:C after 24 h (n = 3). *** indicates *p* < 0.001, ns indicates *p* > 0.05. Abbreviations: µg/mL—micrograms per milliliter, ng/mL—nanograms per milliliter, h—hour, IFN—interferon.

**Figure 4 ijms-25-11872-f004:**
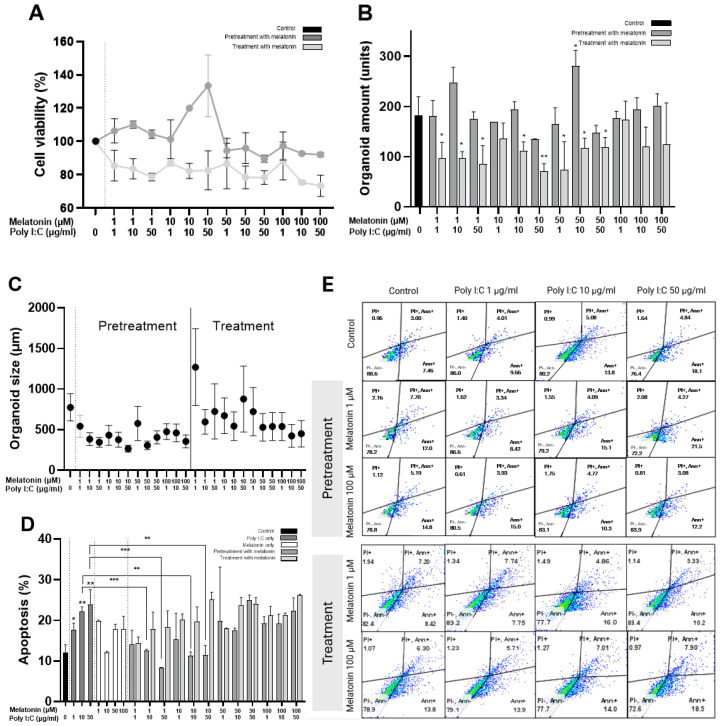
Comparison of the effect of 1, 10, 50 and 100 μM of melatonin before (pretreatment) and after (treatment) stimulation with different concentrations of Poly I:C compared to control cells: (**A**) cell viability (n = 3), (**B**) the number of organoids, (**C**) organoid size, (**D**) apoptosis (n = 3). (**E**) Scatter plots showing the comparison of the effect of 1 and 100 µM of melatonin on apoptosis in Poly I:C-stimulated cells in contrast to the controls. * indicates *p* < 0.05, ** indicates *p* < 0.01, and *** indicates *p* < 0.001. Abbreviations: µg/mL—micrograms per milliliter, µm—micrometer, µM—micromolar.

**Figure 5 ijms-25-11872-f005:**
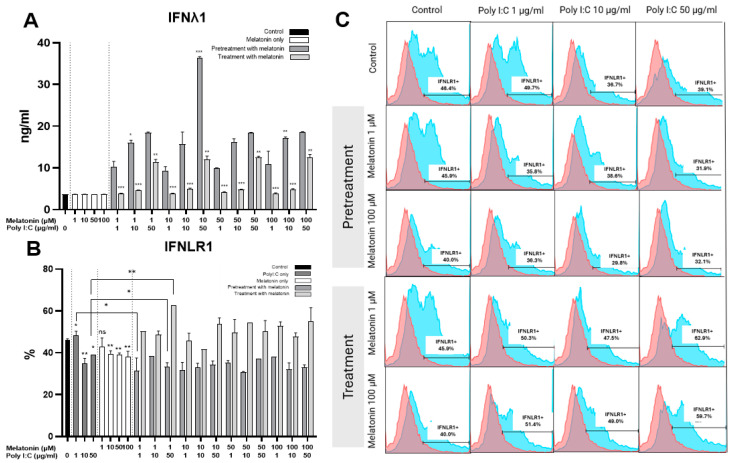
The effect of melatonin on the expression of IFNλ1 and IFNLR1. (**A**) IFNλ1 expression by ELISA in Caco-2 cells treated with different concentrations of melatonin before and after Poly I:C stimulation in comparison to control non-infected cells treated with melatonin (only statistically significant differences in melatonin-pretreated or -treated cells in comparison to melatonin-naïve Poly I:C-stimulated cells are shown) (n = 3). (**B**) IFNLR1 expression by flow cytometry in Caco-2 cells treated with different concentrations of melatonin before and after Poly I:C stimulation in comparison to control non-infected cells treated with melatonin as well as those stimulated by Poly I:C only (n = 3). (**C**) Histograms comparing the effect of 1 and 100 µM of melatonin on the expression of IFNLR1 in Poly I:C-stimulated cells in contrast to the controls (red histogram shows unstained control, blue—IFNLR1 expression). * indicates *p* < 0.05, ** indicates *p* < 0.01, and *** indicates *p* < 0.001, ns indicates *p* > 0.05. Abbreviations: µg/mL—micrograms per milliliter, ng/mL—nanograms per milliliter, µM—micromolar, IFNLR1—IFNλ receptor 1.

**Figure 6 ijms-25-11872-f006:**
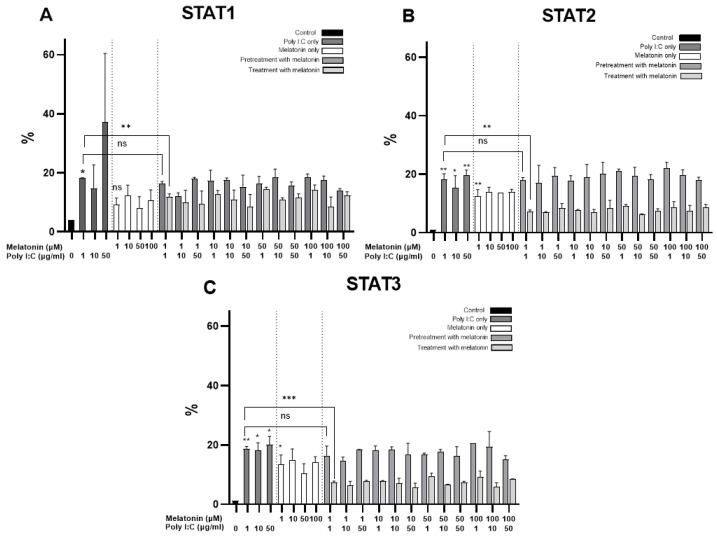
Expression of STAT1-3 proteins by flow cytometry in Caco-2 cells. (**A**) STAT1 expression in control and melatonin-pretreated Caco-2 cells before Poly I:C stimulation in comparison to treated cells (n = 3). (**B**) STAT2 expression in control and melatonin-pretreated Caco-2 cells before Poly I:C stimulation in comparison to treated cells (n = 3). (**C**) STAT3 expression in control and melatonin-pretreated Caco-2 cells before Poly I:C stimulation in comparison to treated cells (n = 3). * indicates *p* < 0.05, ** indicates *p* < 0.01, and *** indicates *p* < 0.001, ns indicates *p* > 0.05. Abbreviations: µg/mL—micrograms per milliliter, µM—micromolar, STAT — signal transducer and activator of transcription.

**Figure 7 ijms-25-11872-f007:**
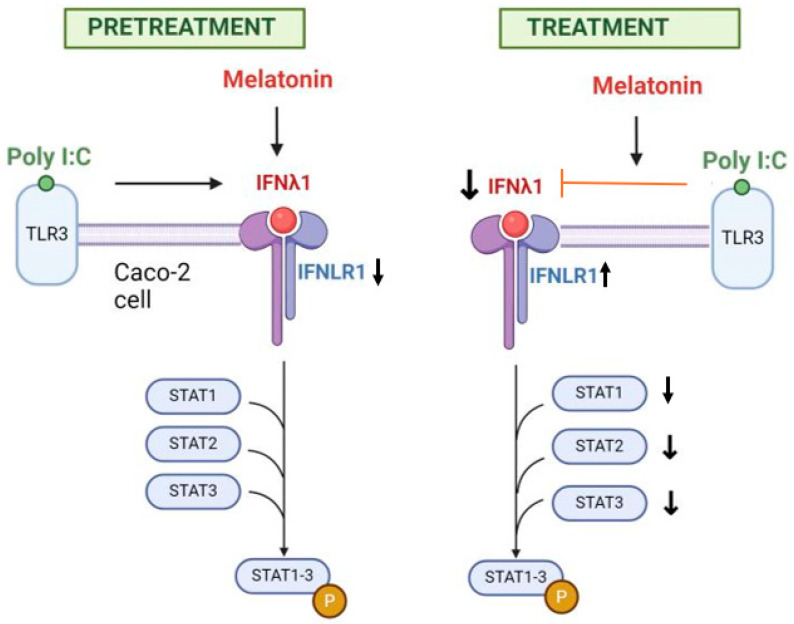
A summary of the research findings. Melatonin, when applied after Poly I:C stimulation, reduces the expression of IFNλ1 and slightly increases the production of IFNLR1 receptor. Furthermore, it reduces the expression of STAT1-3 proteins. In contrast, when used as a pretreatment, it does not have a significant effect on IFNλ1 but reduces IFNLR1 expression, though it does not influence the levels of STAT1-3.

**Figure 8 ijms-25-11872-f008:**
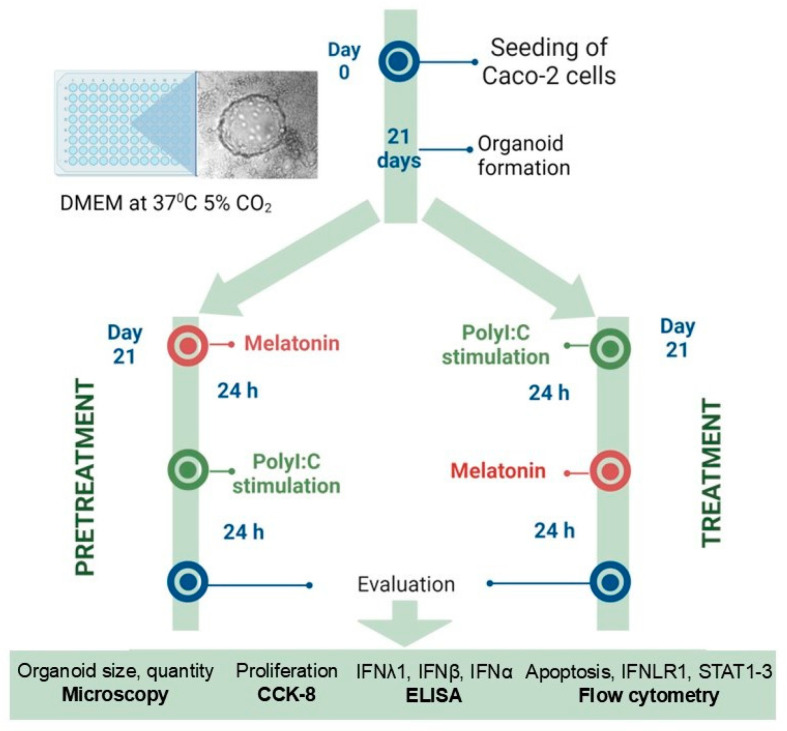
Outline of the research study.

## Data Availability

Data are contained within the article.

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
