# Peer review of "Antiviral Effect of Melatonin on Caco-2 Cell Organoid Culture: Trick or Treat?"

_ijms, 2024, doi:10.3390/ijms252211872_

Round 1
Reviewer 1 Report
Comments and Suggestions for Authors
Minor comments
(Line 12–14): Discuss the primary goal of the study regarding melatonin's effects on Caco-2 cell organoids?
(Line 16–20): Explain in brief melatonin's pretreatment and post-treatment differ in their effects on the organoid size and cell proliferation?
(Line 55–58): Which interferons and important cytokines were examined to determine how melatonin affected viral inflammation?
(Line 61–66): Discuss the Poly I concentrations were utilized to induce viral inflammation in Caco-2 cells, and what effect did this have on the development of organoids and the viability of the cells?
(Line 129–137): What noteworthy discoveries were made about the expression of IFNLR1 and IFNλ1 in melatonin-treated cells upon Poly I stimulation?
(Line 161–169): What function did the STAT proteins (STAT1, STAT2, and STAT3) serve in the investigation, and how were their expression levels impacted by melatonin treatment?
(Line 116–118): what impacts did melatonin pretreatment have on Poly I cell apoptosis levels, and how did these differ from those of post treatment?
(Line 403–407): discuss the findings were made about the relationship between melatonin's antiviral and protective qualities and when and how much of it to take?
Comments on the Quality of English LanguageMinor comments
Author Response
Dear Reviewer 1,
Thank you very much for taking the time to review our manuscript. Please find the detailed responses attached in the file below with the corresponding corrections provided in the file and in track changes in the re-submitted manuscript.

Reviewer 2 Report
Comments and Suggestions for Authors
1. Consider providing data on organoid markers and include 1-2 references to support the claim that Caco-2 cells can grow as organoids.
2. Please clarify whether the Caco-2 "organoids" were harvested/collected from the monolayer and specify the media and incubation conditions used for these collected cells. Additionally, while the organoid model is mentioned in the title and abstract, the discussion (line 186) refers to a monolayer model.
3. To evaluate the antiviral effect of melatonin, an assessment of TLR3/RIG-I/MDA5 is required. Although TLR3 is mentioned in Figure 7, there is no data on its expression.
Author Response
Dear Reviewer 2,
Thank you very much for taking the time to review our manuscript. Please find the detailed responses attached in the file below and the corresponding corrections provided in the file and in track changes in the re-submitted manuscript.

Round 2
Reviewer 2 Report
Comments and Suggestions for Authors
Dear authors, thank you for all the clarifications. I originally thought that you used only spheroids collected from Caco-2 monolayer.
All comments have been properly addressed.